# Biodegradable Polymer Packaging System for ‘Benitaka’ Table Grapes during Cold Storage

**DOI:** 10.3390/polym16020274

**Published:** 2024-01-19

**Authors:** Ricardo Josue Silva, Aline Cristina de Aguiar, Bruno Matheus Simões, Samuel Camilo da Silva, Maíra Tiaki Higuchi, Sergio Ruffo Roberto, Fabio Yamashita

**Affiliations:** Agricultural Research Center, State University of Londrina, Celso Garcia Cid Road, km 380, P.O. Box 10.011, Londrina 86057-970, Brazil; ricardo.josue.silva@uel.br (R.J.S.); alinecristina.aguiar@uel.br (A.C.d.A.); bruno.m.simoes@uel.br (B.M.S.); samuel.camilo.silva@uel.br (S.C.d.S.); maira.tiaki.higuchi@uel.br (M.T.H.); sroberto@uel.br (S.R.R.)

**Keywords:** bio-based liners, cold storage, *Botrytis cinerea*, sulfur dioxide, *Vitis vinifera* L.

## Abstract

A biodegradable polymer packaging system for ‘Benitaka’ table grapes (*Vitis vinifera* L.) was developed to inhibit the development of gray mold during refrigerated storage. The system consisted of packages and sachets containing Na_2_S_2_O_5_ to release sulfur dioxide (SO_2_), both produced with biodegradable films of starch, glycerol, and poly (adipate co-butylene terephthalate) (PBAT) produced via blown extrusion. The films were characterized in terms of thickness, density, mass loss in water, water vapor permeability, sorption isotherms, and mechanical properties. The table grapes were packed with biodegradable plastic bags containing SO_2_-releasing sachets inside. The experimental design was completely randomized, with four repetitions and five treatments: (a) control, without sachet containing Na_2_S_2_O_5_ and SiO_2_; (b) 2 g of Na_2_S_2_O_5_ + 2 g of SiO_2_; (c) 4 g of Na_2_S_2_O_5_ + 1 g of SiO_2_; (d) 4 g of Na_2_S_2_O_5_ + 2 g of SiO_2_; and (e) 4 g of Na_2_S_2_O_5_ + 4 g of SiO_2_. The bunches were stored in a refrigerated chamber at 1 ± 1 °C and relative humidity above 90%. The treatments were evaluated 30 and 45 days after the beginning of refrigerated storage and 3 days at room temperature. The grapes were evaluated based on the incidence of gray mold, mass loss, stem browning, shattered berries, and berry bleaching. The data were subjected to the analysis of variance, and the means were compared using Tukey’s test at 5%. The biodegradable films had good processability during the production via blown extrusion, with good physical properties to be used in the packaging of grapes and the production of SO_2_-releasing sachets. The biodegradable polymer packaging system (biodegradable plastic bags + SO_2_-releasing sachets) inhibited the development of gray mold on ‘Benitaka’ table grapes for 45 days at 1 °C, preserving their quality, with low mass loss, few shattered berries, and rachis freshness.

## 1. Introduction

Sustainable development has now become a major priority across the world. One of the limiting factors of sustainable development is the proliferation of plastic waste [1]. Plastics are fundamentally obtained from petroleum and have been widely used in food packaging due to their low cost, high strength and stability, lightness, and impermeability to gases and numerous solvents, allowing sterilization without affecting food quality [2]. However, the accumulation of large amounts of plastic in the environment is a problem that is growing at an alarming rate. Thus, it is necessary to look for sustainable packaging materials with suitable properties for food packaging [3].

Using biodegradable polymers or biopolymers is a prominent alternative to overcome the consumption of synthetic plastic [4]. The polymers that best adapt to the complete biodegradation process are natural ones, those hydrolyzable to CO_2_ and H_2_O, or CH_4_ and synthetic polymers with structures close to natural ones [5]. Some of the most commercially available synthetic biodegradable polymers are polycaprolactone (PCL), poly (lactic acid) (PLA), poly (adipate co-butylene terephthalate) (PBAT), and poly (adipate co-butylene terephthalate) (PBSA) [4]. These materials have adequate processability for producing plastics with properties comparable to conventional plastics but with high cost, making their use unattractive for the packaging industry.

The main alternative for reducing costs with the use of these polymers is the blend of these synthetic biodegradable polymers with biodegradable polymers from renewable sources such as starch, a macromolecule with low cost, good supply, and with the ability to form polymeric blends [5,6,7]. These blends are compatible with industrial production processes such as extrusion and injection, and can be used to manufacture different packaging and plastic utensils.

Grape is one of the main fruit species cultivated in the world, with a global production of 75 million tons of grapes harvested in 2021, including about 32 million tons of table grapes [8]. The international table grape market demands a high standard of fruit quality, so it is important to maintain the characteristics of the harvested bunches until they reach their final destination [9]. Some elements, such as injuries caused by handling, mass loss, and others caused by pathogens, can compromise the quality of grapes. The main cause of post-harvest losses of table grapes is the fungus *Botrytis cinerea*, the causal agent of gray mold disease [10,11]. Thus, no matter how efficient the phytosanitary treatment is in the field, it should not be dispensed post-harvest [12].

During the refrigerated storage of grapes, sheets that generate sulfur dioxide (SO_2_) are placed inside the packages to control injuries, mainly those caused by *Botrytis cinerea* [13]. Despite being easy to handle and affordable, the sheet selection is not easy, as the gas in high concentration can cause stem browning and the whitening of the berries and harm man and the environment [14,15]. According to Codex Stan 472-2005 regulations, the residual limit of SO_2_ in table grapes cannot exceed 10 ppm or 10 mg/L [16].

To allow table grape conservation for several weeks or even months, the SO_2_ treatment is widely used by packers. Depending on the grapes’ sensitivity to this gas, it can cause damage to the berries, causing fruit bleaching and blocking the product from some markets. Usually, the treatments with SO_2_ and cold storage should kill most arthropod pests and several microorganisms (e.g., the spores of *Botrytis* on the surface of the berry skin). In addition to controlling the development of berry decay, it has an antioxidant action, influencing the physiological processes of the fruit itself, such as maintaining the freshness of the stem [17].

This work aimed to develop a biodegradable active packaging system for ‘Benitaka’ table grapes (*Vitis vinifera* L.) to inhibit the development of gray mold during refrigerated storage.

## 2. Materials and Methods

Two experimental steps were carried out; the first consisted of the development of biodegradable films using a blend of corn starch (Apti, Chapecó, Brazil), glycerol (Dinâmica, Indaiatuba, Brazil) as a plasticizer, and poly (adipate co-butylene terephthalate) (PBAT) (Basf, Ludwigshafen, Germany) produced via blown extrusion. Several formulations were tested, and the one with better mechanical and processability properties was chosen. The second step consisted of the post-harvest conservation of ‘Benitaka’ table grapes (*Vitis vinifera* L.) using biodegradable films and SO_2_-releasing sachets.

### 2.1. Development of Biodegradable Films and SO_2_-Releasing Sachets

#### 2.1.1. Production of Biodegradable Films

Biodegradable films were produced via blown extrusion and, according to preliminary tests, with different concentrations of PBAT (40 to 50%), starch (35 to 45%), and glycerol (15 to 25%). The formulation with better mechanical (highest tensile strength and elongation) and processability properties was 50% of PBAT, 35% of starch, and 15% of glycerol (*w*/*w*/*w*).

The PBAT, starch, and glycerol were mixed and extruded in a single-screw pilot extruder (BGM, model EL-25, Brazil) to produce pellets. According to preliminary tests, the temperature of the four zones of the barrel was set at 90/120/120/125 °C and the screw speed at 35 RPM. The pellets were extruded in the same extruder to produce the films, with the same temperature profile and screw speed.

The dimensions of the biodegradable bags to store the grapes were based on the cardboard box. In this way, it was considered that the size of the bags would be 250 × 350 mm (length × width), enough to avoid mechanical damage to the product and to facilitate sealing when packaging the product. The films were manually cut and sealed using a table sealer.

#### 2.1.2. Production of SO_2_-Releasing Sachets

To produce SO_2_-releasing sachets, four combinations of sodium metabisulphite (Na_2_S_2_O_5_) as an SO_2_-release agent and silica gel beads (SiO_2_) as a moisture control agent were used: 2 g of Na_2_S_2_O_5_ + 2 g of SiO_2_, 4 g of Na_2_S_2_O_5_ + 1 g of SiO_2_, 4 g of Na_2_S_2_O_5_ + 2 g of SiO_2_, and 4 g of Na_2_S_2_O + 4 g of SiO_2_. These combinations permitted varying the amount of SO_2_ released and determining the more efficient formulation.

The 100 × 100 mm sachets were produced with the biodegradable films described in Section 2.1.1, enough to contain sodium metabisulphite and silica gel beads. The biodegradable films were manually cut and sealed with a sealer.

### 2.2. Characterization of Biodegradable Films

#### 2.2.1. Thickness

The thickness was determined using a manual micrometer (±0.001 mm) (Starret, São Paulo, Brazil) at 5 points of the 20 × 50 mm specimen previously conditioned for 48 h at 25 °C and 53% relative humidity (RH). 

#### 2.2.2. Density

The specimens were cut (20 × 20 mm), conditioned in a desiccator containing anhydrous calcium chloride for seven days, weighed, and assessed for mass and geometric area to calculate the density of the material, expressed in g cm^−3^.

#### 2.2.3. Mass Loss in Water (MLW)

The MLW was determined according to the Bortolatto et al. [18] methodology. The specimens were conditioned in a desiccator containing anhydrous calcium chloride for three days, weighed, and immersed in 20 mL of distilled water for 48 h at 25 °C. The specimens were dried at 105 °C in an oven for 4 h and weighed again to determine the weight loss in water (%). The MLW was obtained using Equation (1):(1)$$MLW=\left(\frac{m_{i}-m_{f}}{m_{i}}\right)\times 100$$
where m_i_ is the initial dry mass, and m_f_ is the final dry mass.

#### 2.2.4. Water Vapor Permeability (WVP)

The WVP was determined according to the ASTM E-96-(00) methodology [19]. Measurements were performed in triplicate at a relative humidity (RH) gradient ranging from 33% to 64% (±3%) at 25 °C, using saturated magnesium chloride and sodium nitrite solutions. Before analysis, the specimens were conditioned at 25 °C for 48 h at 53% ± 3% RH.

#### 2.2.5. Water Sorption Isotherms 

The specimens were cut (20 × 20 mm), conditioned in a desiccator containing anhydrous calcium chloride for seven days, and placed in the Aquasorp isotherm generator equipment (Decagon Devices, Pullman, WA, USA). Scanning was carried out in adsorption mode from 0.10 to 0.85 water activity at 25 °C. The sorption isotherms were modeled with the GAB model (Guggenheim, Anderson, and de Boer) using the software Sorptrack 1.14 (Decagon Devices, Pullman, WA, USA), as in Equation (2):(2)$$Xw=\frac{C\cdot K\cdot m_{0}\cdot aw}{\left[\left(1-K\cdot aw\right)\left(1-K\cdot aw+C\cdot K\cdot aw\right)\right]}$$
where Xw is the moisture in dry basis (g water/g solid), aw is the water activity, $m_{0}$ is the molecular monolayer (g water/g solid), C is the constant related to the sorption heat of the molecular layer, and K is the constant related to the sorption heat of the multilayer.

#### 2.2.6. Mechanical Properties 

The mechanical properties of the biodegradable films were determined according to the ASTM D882-02 [20] methodology using a TA.XT2i Plus texturometer (Stable Micro Systems, Godalming, UK). The film specimens (50 mm × 20 mm) were fixed to the equipment’s grips with an initial distance of 30 mm and a speed of 0.8 mm s^−1^. The determined properties were the maximum tensile strength, elongation at break, and Young’s Modulus.

### 2.3. Post-Harvest Conservation of ‘Benitaka’ Table Grapes Using Biodegradable Packaging and SO_2_-Releasing Sachets 

#### 2.3.1. Description of Treatments 

Fresh bunches of the ‘Benitaka’ table grapes (*Vitis vinifera* L.) were harvested from a commercial vineyard located in Cambira, PR, Brazil (23°35′ S, 51°34′ W, elevation of 1017 m), with a history of occurrence of *Botrytis cinerea*. The region is classified as subtropical (Cfa) according to Köppen, with an average annual temperature of 20.7 °C and annual precipitation of 1640 mm [21]. The harvest was carried out during the 2020 summer season when the soluble solids content (SST) of the grapes reached around 14° Brix. 

The grapes were packed in biodegradable plastic bags without perforation with an SO_2_-releasing sachet containing sodium metabisulphite (Na_2_S_2_O_5_) as an active ingredient and silica pearls (SiO_2_) as a moisture regulator. The experimental design used was completely randomized, with five treatments and four replications, as follows: (a) control, without sachet; (b) sachets with 2 g of Na_2_S_2_O_5_ + 2 g of SiO_2_; (c) sachets with 4 g of Na_2_S_2_O_5_ + 1 g of SiO_2_; (d) sachets with 4 g of Na_2_S_2_O_5_ + 2 g of SiO_2_; and (e) sachets with 4 g of Na_2_S_2_O_5_ + 4 g of SiO_2_.

#### 2.3.2. Grapes Packaging

The grape bunches were cleaned and standardized according to their appearance and mass, and ~2 kg was placed in biodegradable plastic bags and packaged according to the treatment (Figure 1). Then, the corrugated cardboard boxes were placed in a refrigerated chamber at 1 ± 1 °C with 90% RH for 45 days.

#### 2.3.3. Assessments

The grapes were evaluated 30 and 45 days after the beginning of refrigerated storage, and the following variables were analyzed: incidence of gray mold on the berries, bunch mass loss, stem browning, shattered berries, and berry bleaching. After 45 days at 1 ± 1 °C, the packed grapes were stored for 3 days at room temperature (22 ± 1 °C) and re-evaluated, except for bunch mass loss.

The incidence of gray rot in the berries was quantified by [22]: incidence (%) = (number of infected berries/total number of berries in the bunch) × 100. The bunch mass loss was determined via weighing the bunches at the beginning of storage and at the time of each evaluation, according to Mattiuz et al. [23]: mass loss (%) = [(initial mass − mass at the time of assessment)/(initial mass)] × 100. The stem browning was evaluated with visual evaluation according to the methodology described by Ngcobo et al. [24], assigning scores according to the browning level: 1 (fresh and green), 2 (slightly brown), 3 (significantly brown), and 4 (severely brown). Shattered berries were evaluated via counting the loose berries and expressed as a percentage. Berry bleaching was quantified according to the formula described by Henríquez et al. [25]: berry bleaching (%) = (number of berries with bleaching/total number of berries in the bunch) × 100.

### 2.4. Statistical Analysis

The experimental data presented a normal distribution according to the Shapiro–Wilk test. Then the data were subjected to the analysis of variance (ANOVA), and the treatment means were compared using Tukey’s test at 5% significance (*p* < 0.05). The statistical analyses were performed using the Statistica software 7.0^®^ (StatSoft, Street Tulsa, OK, USA).

## 3. Results and Discussion

### 3.1. Characteristics of the Biodegradable Film

The physical properties of the biodegradable film are presented in Table 1, Table 2 and Table 3.

The thickness of the biodegradable film was 111 µm, and its density was 1.22 g cm^−3^ (Table 1). The higher the film density, the higher the amount of material used to produce them, increasing the material cost. Kormin et al. [26] produced injected low-density polyethylene (LDPE) materials (density of 0.9188 g cm^−3^) with two different starches and evaluated the densities of these materials. According to the authors, the density of the materials increased with an increasing proportion of starch in the mixture, probably due to the weak binding between starch and LDPE, as thermoplastic starch is hydrophilic. LDPE is hydrophobic; the same behavior can be suggested for films with PBAT and starch. 

Brandelero, Grossmann, and Yamashita [27] produced films via blown extrusion, and the control formulation had a composition similar to that of this study (thermoplastic starch/PBAT 65/35 m/m). The average thickness was 200 µm, and the average density was 1.34 g cm^−3^, higher than reported in the present work, probably due to the higher thermoplastic starch content, 65% against 50% of our formulation.

The biodegradable film used for packaging grapes was relatively stable in water since the MLW was 15.9%. This is an excellent outcome since starch-based materials are highly hygroscopic and, consequently, have a high MLW [28].

The biodegradable film’s water vapor permeability (WVP) was about 6.6 × 10^−11^ g m^−1^ s^−1^ Pa^−1^; therefore, the film can be considered highly permeable to water vapor. In other studies, with films formulated with PBAT and starch in proportions similar to the present work, Müller, Yamashita, and Laurindo [29] reported WVP of 9.5 × 10^−11^ g m^−1^ s^−1^ Pa^−1^ and Dias [30] of 5.2 × 10^−11^g m^−1^ s^−1^ Pa^−1^, that is, similar to those determined in the present work. Brandelero, Grossmann, and Yamashita [27], for films of thermoplastic starch and PBAT containing Tween 80 and soybean oil in the matrix, reported WVP of 2.8 × 10^−12^ g m^−1^ s^−1^ Pa^−1^, for a gradient of 32–64% RH, this difference being due to the presence of oil in the matrix, making the material more hydrophobic.

The GAB model parameters for the water sorption isotherm of the biodegradable film are shown in Table 2. The model adjustment to the experimental data was excellent, with a coefficient of determination (R^2^) close to 1 and *m*_0_ equal to 10.7 g water/100 g solids. Costa [31] reported an *m*_0_ of 7.3 g water/100 g solids for a biodegradable film of thermoplastic starch/PBAT (70/30 *w*/*w*). Müller, Yamashita, and Laurindo [29], for a film of 100% thermoplastic starch containing 30% glycerol, obtained an *m*_0_ of 9.4 g water/100 g solids. 

The biodegradable film’s tensile strength was 8.8 MPa, the elongation was 604%, and the Young’s Modulus was 45.0 MPa (Table 3). According to studies carried out by Brandelero, Grossmann, and Yamashita [27], the biodegradable films produced via blown extrusion with thermoplastic starch and PBAT, with proportions similar to those used in this work, had a tensile strength of 4 MPa and an elongation at break of 50%. Olivato et al. [32] reported values of 6 MPa for tensile strength, elongation of 150%, and Young’s Modulus of 55 MPa for thermoplastic starch/PBAT (55/45 *w*/*w*) films produced via blown extrusion.

The developed biodegradable film is a good option for application as biodegradable packaging because of its physical properties.

### 3.2. Post-Harvest Conservation of ‘Benitaka’ Table Grapes Using Biodegradable Packaging and SO_2_-Releasing Sachets

#### 3.2.1. Incidence of Gray Mold on Berries

After 30 and 45 days of storage in a refrigerated chamber at 1 ± 1 °C, ‘Benitaka’ grapes packaged in biodegradable plastic bags containing SO_2_-releasing sachets had a lower incidence or even absence of gray mold on the berries, statistically differing from the treatment control, without sachet (Table 4), due to the SO_2_ released inside the packaging.

The high hydrophilicity of silica gel and the high WVP of the biodegradable films controlled the release of SO_2_ during storage, as they balanced the water content to react with the metabisulfite and release SO_2_. There was no difference between treatments because the released amount of SO_2_ from all sachets was enough to control the development of *Botrytis cinerea*.

Furthermore, the bags retained the SO_2,_ creating a suitable micro atmosphere for the release and action of SO_2_ without causing physiological damage to the grapes due to the CO_2_ accumulation or lack of O_2_, i.e., the biodegradable plastic bags plus the sachets acted as a modified atmosphere packaging and as an active packaging system.

After 3 days at room temperature (22 °C), in which all grapes were under the same condition, i.e., out of the bags, the low incidence or absence of the gray mold on the berries previously packed with SO_2_-releasing sachets confirmed the eradicating effect of the SO_2_. In bunches treated with SO_2_-releasing sachets, the disease incidence ranged from 0.0 to 0.34% of berries with symptoms, differing from the control treatment, which had 7.15% of infected berries.

The incidence of gray mold (% of diseased berries) was lower than those reported by Aguiar and Higuchi et al. [33], who worked with bio-based, laser-perforated, and recyclable SO_2_-generating liners, alone or in combination with ultra-fast SO_2_-generating before packaging, to extend the shelf life of ‘Benitaka’ table grapes (*Vitis vinifera* L.). The authors reported 0 to 1% after 30 days at 1 °C, 0.2 to 2.1% after 45 days at 1 °C, and 0.1 to 3.5% after 3 days at room temperature without packaging after 45 days at 1 °C.

According to Smilanick et al. and Chervin et al. [34,35], *Botrytis cinerea* infection often remains latent, and grapes must be continuously exposed to the gas to control the disease via periodically eliminating the growing mycelium. However, for the grapes packed with SO_2_-releasing sachets, the incidence of rot after removal from the packaging and kept for 3 days at room temperature was very low or non-existent, i.e., the use of sachets prevented the germination of active spores due to the antifungal action of SO_2_.

Regulation N° 543/2011 of the European Union (EU) [36] establishes rules for the import of fruits and vegetables. In Annex 1 of the regulation, the United Nations Economic Commission for Europe (UNECE) establishes the trade and quality control rules for exporting fresh grapes to the EU [37]. This regulation establishes that fresh grapes may contain light surface defects and must not show signs of rotting or deterioration that make them unfit for consumption.

#### 3.2.2. Mass Loss

There was no statistical difference between treatments for the mass loss of the grapes after 30 days and 45 days of cold storage (Table 5) because the same package was used for all treatments, including the control, but the longer the storage period, the higher the grapes’ mass loss. The packaging used in this trial controlled the transmission of water vapor, and the lower the transmission rate, the greater the relative humidity inside the packaging, leading to the reduction in transpiration and, consequently, mass loss [38,39].

After 45 days of storage at 1 °C, the grapes had a mass loss that ranged from 4.62 to 5.82, similar to those reported by Aguiar and Higuchi et al. [33]. According to Gorgatti-Netto [40], when the mass loss of grapes reaches about 4 to 5%, it affects the ideal appearance and firmness for consumption. Excessive mass loss can cause stem browning and soften the berries, facilitating the shattered berries [41,42,43].

#### 3.2.3. Stem Browning

After 30 days of refrigerated storage at 1 °C, the stem browning had no statistical difference between the treatments (Table 6). Still, after 45 days, the lowest browning scores were observed for the treatments that used SO_2_-releasing sachets. These results were similar to those reported by Aguiar and Higuchi et al. [33].

After 3 days at room temperature, without the biodegradable plastic packaging, the stem browning of the control treatment was higher than the treatments containing SO_2_-releasing sachets, losing its fresh appearance (score > 3). According to Ahmed et al. [44], after 50 days of refrigerated storage, the bunches of ‘Italia’ grapes packed with SO_2_ dual-phase release sheets had lower stem browning than the treatment consisting only of the microperforated plastic bag. The SO_2_ gas has the ability to control the decay, but also influences the physiological characteristics of the bunches, such as the maintenance of the freshness of the stems and the turgidity of the berries [42], due to the inhibitory action of SO_2_ on the catalytic mechanism of some enzymes that favor the respiration process [40].

Despite the high browning score, the appearance of the rachis after 3 days at 22 °C was acceptable, as only intense browning can harm commercialization [43].

#### 3.2.4. Shattered Berries

The percentage of shattered berries after 30 and 45 days at 1 °C and 3 days at 22 °C did not differ between the treatments (Table 7) and ranged from 0.34 to 5.44%, i.e., the SO_2_-releasing sachets did not affect the shedding compared to the control treatment. These results were similar to those reported by Aguiar and Higuchi et al. [33]. According to the technical regulation of identity and quality for classifying fine table grapes (Annex 2 of Normative Instruction N° 001), the limits accepted for berry shatter are, at maximum, 5% [45].

Despite being considered a defect in grapes bunches, the shattered berries can often go unnoticed by the consumer. In bunches of the ‘Niagara Rosada’ table grape, evaluated according to the visual appearance, 0, 5, 10, and 15% of shattered berries did not differ in consumer preference [46].

#### 3.2.5. Berry Bleaching

Grapes treated with SO_2_-releasing sachets resulted in lesions due to the bleaching of the berries (Table 8), and there was no statistical difference between the treatments with SO_2_-releasing sachets. The berries of the control treatment did not show bleaching.

The percentage of berries with bleaching over time increased, even after 3 days at 22 °C when the biodegradable plastic packaging and the SO_2_-releasing sachet were no longer used, and ranged from 20.94 to 33.43%. 

The generation of H_2_SO_3_ and H_2_SO_4_ acids via SO_2_ after contact with water vapor can induce the bleaching of berries [46]. When only the color of the berries is affected, bleaching is classified as a mild defect due to SO_2_ injury, with a maximum limit of 12%. However, the affected berry area can become soft as this defect progresses and sometimes cracks appear. At this stage, bleaching is classified as a severe defect due to SO_2_ damage, with a maximum limit of 4% [47].

The bunches of ‘Benitaka’ grapes submitted to treatments with SO_2_-releasing sachets after 45 days at 1 °C showed levels above the acceptable limit of bleaching, between 15.32 and 22.13% and after 3 days at room temperature, levels of bleaching between 20.94 and 33.43%. Therefore, reducing the amount of active ingredient in the sachets, increasing the SO_2_ permeability of the biodegradable plastic bags using microperforations, reducing the material thickness, or changing the material formulations can solve this problem.

## 4. Conclusions

The biodegradable polymers had good processability during the blown extrusion production process because they maintained a continuous production flow and ensured uniform film thickness. These polymer films have good physical properties and can be used for packaging grapes and producing SO_2_-releasing sachets. The biodegradable active packaging (biodegradable plastic bags + SO_2_-releasing sachets) inhibited the development of gray mold on ‘Benitaka’ table grapes for 45 days at 1 °C, preserving their quality, with low mass loss, few shattered berries, and rachis freshness, but resulted in lesions due to the bleaching of the berries. It is necessary to test lower amounts of sodium metabisulphite in the sachets. Using biodegradable packaging, even if only partially replacing traditional plastic options, can significantly lessen the negative impact on our environment. However, it is necessary to reduce the material costs, as biodegradable polymers such as PBAT are still more expensive than conventional polymers such as polyethylene and polypropylenes.

## Figures and Tables

**Figure 1 polymers-16-00274-f001:**
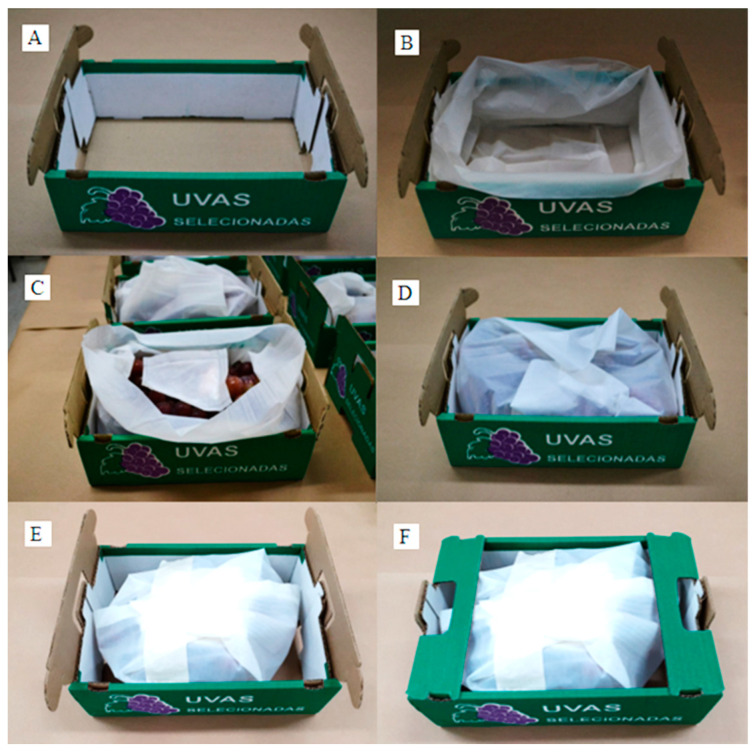
Steps of grapes’ packing. (**A**) Accommodation of the biodegradable plastic bag into the cardboard box; (**B**) Accommodation of the bunches inside the bag; (**C**) Placing the SO_2_-releasing sachet over the grapes inside the biodegradable plastic bag; (**D**) Folding of the biodegradable plastic bag; (**E**) Closing with a seal; (**F**) Closing the box for storage in the chamber.

**Table 1 polymers-16-00274-t001:** Thickness, density, mass loss in water (MLW), and water vapor permeability (WVP) of the biodegradable film.

Thickness (µm)	Density (g cm^−3^)	MLW (%)	WVP (×10^11^) (g m^−1^ s^−1^ Pa^−1^)
111 ± 7	1.223 ± 0.020	15.9 ± 0.2	6.6 ± 0.2

Results are expressed as mean (±standard deviation).

**Table 2 polymers-16-00274-t002:** Water sorption isotherms (GAB model parameters) of the biodegradable film.

GAB Model Parameters			
*C*	*K*	*m*_0_ (g Water/100 g Solids)	R^2^
15.3	0.61	10.7	0.99

**Table 3 polymers-16-00274-t003:** Mechanical properties of the biodegradable film.

Mechanical Properties		
Tensile Strength (MPa)	Elongation at Break (%)	Young’s Modulus (MPa)
8.8 ± 0.3	604 ± 32	45.0 ± 0.9

Results are expressed as mean (±standard deviation).

**Table 4 polymers-16-00274-t004:** Incidence of gray mold in ‘Benitaka’ table grapes after 30 and 45 days of storage in a refrigerated chamber at 1 °C, followed by 3 days at room temperature (22 ± 1 °C), packed in bulk in cardboard boxes and biodegradable plastic bags with different SO_2_-releasing sachets.

Treatment	Incidence of Gray Mold (% of Diseased Berries)		
	30 Days at 1 °C	45 Days at 1 °C	3 Days at 22 °C without Packaging after 45 Days at 1 °C
Control	2.38 ± 0.19 b	4.60 ± 0.51 b	7.15 ± 1.58 b
2 g of Na_2_S_2_O_5_ + 2 g of SiO_2_	0.00 ± 0.00 a	0.00 ± 0.00 a	0.00 ± 0.00 a
4 g of Na_2_S_2_O_5_ + 1 g of SiO_2_	0.00 ± 0.00 a	0.00 ± 0.00 a	0.00 ± 0.00 a
4 g of Na_2_S_2_O_5_ + 2 g of SiO_2_	0.17 ± 0.17 a	0.17 ± 0.17 a	0.34 ± 0.34 a
4 g of Na_2_S_2_O_5_ + 4 g of SiO_2_	0.00 ± 0.00 a	0.00 ± 0.00 a	0.17 ± 0.17 a

Results are expressed as mean (±standard deviation). Means followed by the same letter in the columns do not differ statistically according to Tukey’s test (*p* ≤ 0.05).

**Table 5 polymers-16-00274-t005:** Mass loss (%) of ‘Benitaka’ table grape bunches after 30 and 45 days of storage at 1 °C, packed in bulk in cardboard boxes and biodegradable plastic bags with different SO_2_-releasing sachets.

Treatment	Mass Loss (%)	
	30 Days at 1 °C	45 Days at 1 °C
Control	3.37 ± 0.30 a	4.62 ± 0.81 a
2 g of Na_2_S_2_O_5_ + 2 g of SiO_2_	3.42 ± 0.47 a	4.75 ± 0.45 a
4 g of Na_2_S_2_O_5_ + 1 g of SiO_2_	3.52 ± 0.80 a	4.95 ± 0.80 a
4 g of Na_2_S_2_O_5_ + 2 g of SiO_2_	3.85 ± 0.24 a	5.20 ± 0.28 a
4 g of Na_2_S_2_O_5_ + 4 g of SiO_2_	4.20 ± 1.17 a	5.82 ± 0.97 a

Results are expressed as mean (±standard deviation). Means followed by the same letter in the columns do not differ statistically according to Tukey’s test (*p* ≤ 0.05).

**Table 6 polymers-16-00274-t006:** Stem browning of bunches of ‘Benitaka’ table grapes after 30 and 45 days at 1 °C, followed by 3 days at room temperature at 22 °C, packed in bulk in cardboard boxes and biodegradable plastic bags with different SO_2_-releasing sachets.

Treatment	Stem Browning *		
	30 Days at 1 °C	45 Days at 1 °C	3 Days at 22 °C without Packaging after 45 Days at 1 °C
Control	1.00 ± 0.00 a	2.00 ± 0.00 b	3.25 ± 0.25 b
2 g of Na_2_S_2_O_5_ + 2 g of SiO_2_	1.00 ± 0.00 a	1.00 ± 0.00 a	1.50 ± 0.28 a
4 g of Na_2_S_2_O_5_ + 1 g of SiO_2_	1.00 ± 0.00 a	1.00 ± 0.45 a	1.25 ± 0.25 a
4 g of Na_2_S_2_O_5_ + 2 g of SiO_2_	1.50 ± 0.28 a	1.50 ± 0.58 a	1.50 ± 0.28 a
4 g of Na_2_S_2_O_5_ + 4 g of SiO_2_	1.25 ± 0.25 a	1.25 ± 0.45 a	1.25 ± 0.25 a

* Score of stem browning: 1—fresh and green; 2—slightly brown; 3—significantly brown; and 4—severely brown [24]. Results are expressed as mean (±standard deviation). Means followed by the same letter in the columns do not differ statistically according to Tukey’s test (*p* ≤ 0.05).

**Table 7 polymers-16-00274-t007:** Percentage of shattered berries from ‘Benitaka’ table grapes after 30 and 45 days at 1 °C, followed by 3 days at 22 °C, packed in bulk in cardboard boxes and biodegradable plastic bags with different SO_2_-releasing sachets.

Treatment	Shattered Berries (%)		
	30 Days at 1 °C	45 Days at 1 °C	3 Days at 22 °C without Packaging after 45 Days at 1 °C
Control	0.34 ± 0.19 a	1.87 ± 0.42 a	2.38 ± 0.44 a
2 g of Na_2_S_2_O_5_ + 2 g of SiO_2_	1.19 ± 0.51 a	2.55 ± 1.05 a	5.44 ± 2.20 a
4 g of Na_2_S_2_O_5_ + 1 g of SiO_2_	0.68 ± 0.00 a	1.70 ± 0.43 a	3.23 ± 0.17 a
4 g of Na_2_S_2_O_5_ + 2 g of SiO_2_	1.02 ± 0.43 a	1.87 ± 0.75 a	4.25 ± 0.97 a
4 g of Na_2_S_2_O_5_ + 4 g of SiO_2_	0.51 ± 0.17 a	2.72 ± 0.83 a	5.10 ± 2.18 a

Results are expressed as mean (±standard deviation). Means followed by the same letter in the columns do not differ statistically according to Tukey’s test (*p* ≤ 0.05).

**Table 8 polymers-16-00274-t008:** Percentage of berry bleaching of ‘Benitaka’ table grapes after 30 and 45 days at 1 °C, followed by 3 days at room temperature at 22 °C, packed in bulk in cardboard boxes and biodegradable plastic bags with different SO_2_-releasing sachets.

Treatment	Berry Bleaching (%)		
	30 Days at 1 °C	45 Days at 1 °C	3 Days at 22 °C without Packaging after 45 Days at 1 °C
Control	0.00 ± 0.00 b	0.00 ± 0.00 b	0.00 ± 0.00 b
2 g of Na_2_S_2_O_5_ + 2 g of SiO_2_	5.62 ± 1.12 a	17.37 ± 2.68 a	20.94 ± 1.87 a
4 g of Na_2_S_2_O_5_ + 1 g of SiO_2_	4.94 ± 0.51 a	22.13 ± 3.10 a	33.43 ± 3.55 a
4 g of Na_2_S_2_O_5_ + 2 g of SiO_2_	6.64 ± 0.93 a	17.54 ± 3.61 a	25.88 ± 8.37 a
4 g of Na_2_S_2_O_5_ + 4 g of SiO_2_	5.61 ± 1.95 a	15.32 ± 2.26 a	27.76 ± 2.12 a

Results are expressed as mean (±standard deviation). Means followed by the same letter in the columns do not differ statistically according to Tukey’s test (*p* ≤ 0.05).

## Data Availability

All the data analyzed in the study are included in the tables and figures in this published article.

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
