# Peer review of "Biodegradable Polymer Packaging System for ‘Benitaka’ Table Grapes during Cold Storage"

_polymers, 2024, doi:10.3390/polym16020274_

Round 1
Reviewer 1 Report
Comments and Suggestions for Authors
This article discusses the biodegradable packaging system for ‘Benitaka’ table grapes during cold refrigeration. I found this work interesting and scientifically sound. Therefore, I would recommend this paper for publication after addressing the following comments.
· The novelty of this work needs to be better elaborated.
· There are numerous typos and punctuation errors in the text. Please carefully go through and correct them.
· Please compare the results of previous literature that are similar to the current study.
For instance, the paper, DOI:10.1016/j.postharvbio.2022.112217 has not been discussed.
· Why is the berry bleaching much higher for treated samples than for control samples?
· Please insert the images of the samples to see the actual defects.
· Characterization of the packaging materials is also essential. Hence, describe how the packaging materials were characterized.
Comments on the Quality of English Language
English needs to be improved.
Author Response
Response to Reviewers
Dear Reviewers, first of all, thank you very much for your comments and remarks. We tried to address all of them, and this new version of our manuscript was changed and improved according to your recommendation, as it can be checked below:
Reviewers’ comments:
Reviewer #1:
This article discusses the biodegradable packaging system for ‘Benitaka’ table grapes during cold refrigeration. I found this work interesting and scientifically sound. Therefore, I would recommend this paper for publication after addressing the following comments.
The novelty of this work needs to be better elaborated.
There are numerous typos and punctuation errors in the text. Please carefully go through and correct them.
R: We have corrected the text to improve all these comments and suggestions.
Please compare the results of previous literature that are similar to the current study. For instance, the paper, DOI:10.1016/j.postharvbio.2022.112217 has not been discussed.
R: We include the comparison between the 2 works in the discussion.
Why is the berry bleaching much higher for treated samples than for control samples?
R: The reason that bleaching is much higher for treated samples is because of the presence of the Sodium Metabisulfite (Na2S2O5) in these treatments, while in control. Depending the concentration of this ingredient and the sensibility of the grape cultivar, the incidence of bleaching can vary, and in the case of the control, as no chemical was used, no bleaching was observed on berries.
Please insert the images of the samples to see the actual defects.
R: Unfortunately, we do not have such images to be shown the reviewer mentioned.
Characterization of the packaging materials is also essential. Hence, describe how the packaging materials were characterized.
R: We included more information about the characterization.
Reviewer 2 Report
Comments and Suggestions for Authors
Thank you very much for the opportunity to review the article “Biodegradable polymer packaging system for ‘Benitaka’ table grape during cold storage.” The article presents the results of studies on the storage of 'Benitaka' grapes (Vitis vinifera L.) using biodegradable packaging based on starch/adipate co-butylene terephthalate (PBAT)/glycerol, as well as sodium metabisulphite (Na2S2O5) as a preservative and silica gel ( SiO2) as a moisture-regulating additive.
Abstract is written very intriguingly, it stirs up interest. The Introduction section is written very logically and interconnectedly. A very short text contains a large amount of information, which is very competently connected with all aspects of the presented research, and a rather interesting and ambitious goal is formulated. And the reader begins to imagine a large amount of experimental data with photographs and graphs that await him in the following sections. But then something went wrong.
The Materials and Methods section, lines 85 - 91, talks about two stages of the experiment, but the Results and Discussion section presents the results of only one stage. The results of the experiments described in subsection “2.1.1. Production of biodegradable films” are not presented. And this is a very large multifactorial experiment. If these data were published earlier, then you should describe the composition you have chosen and provide a link to the source that describes these studies. Then the materials and methods section should be cleared of methods that you did not use. Or you should give the results of these studies; the reader is interested in knowing how you chose the optimal composition.
Line 127 – 128. All modifications to the method must be described and justified.
Line 170 – 171. Capacity 2 kg, is this the capabilities of the package, or a bunch of grapes in the experiment?
Tables 1, 2, 3 create extra volume in this form. Here you need to either present all the results, or a brief description of the material and its properties, indicating a link to the source where you can get acquainted with more detailed results of the experiments.
How was it determined that Botrytis cinerea infection had occurred? How was the pathogen identified? We need at least photographs of the infestations.
Table 6 is not clear. If a visual assessment implies integer point values, 1, 2, 3, 4, how to interpret fractional values in the table? How were these values obtained? Materials and Methods should include a photograph of the color chart.
Very few results and a lot of discussion. Only the results of the analysis are presented, all primary information is hidden. Presenting material in the form of tables is not always justified; a large volume is collected. Many tables could be presented in the form of graphs, histograms and photographs, combined into blocks.
The article needs a lot of work.
Author Response
Response to Reviewers
Dear Reviewers, first of all, thank you very much for your comments and remarks. We tried to address all of them, and this new version of our manuscript was changed and improved according to your recommendation, as it can be checked below:
Reviewers’ comments:
Reviewer #2:
Thank you very much for the opportunity to review the article “Biodegradable polymer packaging system for ‘Benitaka’ table grape during cold storage.” The article presents the results of studies on the storage of 'Benitaka' grapes (Vitis vinifera L.) using biodegradable packaging based on starch/adipate co-butylene terephthalate (PBAT)/glycerol, as well as sodium metabisulphite (Na2S2O5) as a preservative and silica gel ( SiO2) as a moisture-regulating additive.
Abstract is written very intriguingly, it stirs up interest. The Introduction section is written very logically and interconnectedly. A very short text contains a large amount of information, which is very competently connected with all aspects of the presented research, and a rather interesting and ambitious goal is formulated. And the reader begins to imagine a large amount of experimental data with photographs and graphs that await him in the following sections. But then something went wrong.
The Materials and Methods section, lines 85 - 91, talks about two stages of the experiment, but the Results and Discussion section presents the results of only one stage.
R: The results of the first step were not shown because it was a previous test to determine the best film formulation (concentration of starch, PBAT, and glycerol), i.e., the film with better mechanical and processability properties was chosen. We included the information “data not shown” to clarify.
The results of the experiments described in subsection “2.1.1. Production of biodegradable films” are not presented. And this is a very large multifactorial experiment. If these data were published earlier, then you should describe the composition you have chosen and provide a link to the source that describes these studies. Then the materials and methods section should be cleared of methods that you did not use. Or you should give the results of these studies; the reader is interested in knowing how you chose the optimal composition.
R: Due to the size limitation of the article and the objective of the work, we decided not to include the results of the mechanical properties of the films that were not used to package the grapes. We included the information in lines 96-97 that we chose the film with the highest tensile strength and elongation. All the methods described in the Materials and Methods section were used.
Line 127 – 128. All modifications to the method must be described and justified.
R: The reference was wrong. The correct reference has been placed in the text.
Line 170 – 171. Capacity 2 kg, is this the capabilities of the package, or a bunch of grapes in the experiment?
R: Each package had ~2 kg of grapes. We modified the text to clarify.
Tables 1, 2, 3 create extra volume in this form. Here you need to either present all the results, or a brief description of the material and its properties, indicating a link to the source where you can get acquainted with more detailed results of the experiments.
R: Dear reviewer, we understand you point of view, but our group discussed it and we decided to maintain all the Tables in this current form to better show the results. Changes like this would completely remodel the article, and we believe that, as in other publications of ours, this format is sufficient to readers understand the results.
How was it determined that Botrytis cinerea infection had occurred? How was the pathogen identified? We need at least photographs of the infestations.
R: Gray mold caused by Botrytis cinerea is widely known as the main pathogen in table grapes during cold storage. For this reason, no image was shown along the manuscript because it wouldn’t bring any novelty. The incidence of this disease was assessed as the % of berries with decay. As we have a large experience on identifying this disease, a visual assessment was taken to determine the disease incidence.
Table 6 is not clear. If a visual assessment implies integer point values, 1, 2, 3, 4, how to interpret fractional values in the table? How were these values obtained? Materials and Methods should include a photograph of the color chart.
R: Dear reviewer, the values of stem browning scores were obtained by using a visual assessment according to the reference: Ngcobo, M.E.K.; Opara, U.L.; Thiart, G.D. Effects of packaging liners on cooling rate and quality attributes of table grape (cv. Regal Seedless). Packag. Technol. Sci. 2011, 25, 73-84. This scoring system is widely known in the literature, and several authors have already cited. Our group has already presented this image at: Domingues, A.R.; Roberto, S.R.; Ahmed, S.; Shahab, M.; José Chaves Junior, O.; Sumida, C.H.; De Souza, R.T. Postharvest Techniques to Prevent the Incidence of Botrytis Mold of ‘BRS Vitoria’ Seedless Grape under Cold Storage. Horticulturae 2018, 4, 17. https://doi.org/10.3390/horticulturae4030017, and for this reason, we believe that to show it again in this work would be unnecessary.
Round 2
Reviewer 1 Report
Comments and Suggestions for Authors
Satisfied with the changes and revisions made, hence recommend for publication in the current form.
Reviewer 2 Report
Comments and Suggestions for Authors
Dear authors, thank you for your explanations.